# Complex Structure of *Lasiopodomys mandarinus vinogradovi* Sex Chromosomes, Sex Determination, and Intraspecific Autosomal Polymorphism

**DOI:** 10.3390/genes11040374

**Published:** 2020-03-30

**Authors:** Svetlana A. Romanenko, Antonina V. Smorkatcheva, Yulia M. Kovalskaya, Dmitry Yu. Prokopov, Natalya A. Lemskaya, Olga L. Gladkikh, Ivan A. Polikarpov, Natalia A. Serdyukova, Vladimir A. Trifonov, Anna S. Molodtseva, Patricia C. M. O’Brien, Feodor N. Golenishchev, Malcolm A. Ferguson-Smith, Alexander S. Graphodatsky

**Affiliations:** 1Institute of Molecular and Cellular Biology, Siberian Branch of Russian Academy of Sciences, Novosibirsk 630090, Russia; dprokopov@mcb.nsc.ru (D.Y.P.); lemnat@mcb.nsc.ru (N.A.L.); olga_gladkikh@mcb.nsc.ru (O.L.G.); serd@mcb.nsc.ru (N.A.S.); vlad@mcb.nsc.ru (V.A.T.); rada@mcb.nsc.ru (A.S.M.); graf@mcb.nsc.ru (A.S.G.); 2Department of Vertebrate Zoology, St. Petersburg State University, St. Petersburg 199034, Russia; tonyas1965@mail.ru (A.V.S.); ivanApolikarpov@gmail.com (I.A.P.); 3Severtzov Institute of Ecology and Evolution, Russian Academy of Sciences, Moscow 119071, Russia; sicistam@yandex.ru; 4Department of Natural Science, Novosibirsk State University, Novosibirsk 630090, Russia; 5Cambridge Resource Centre for Comparative Genomics, Department of Veterinary Medicine, University of Cambridge, Madingley Road, Cambridge CB3 OES, UKmaf12@cam.ac.uk (M.A.F.-S.); 6Zoological Institute, Russian Academy of Sciences, Saint-Petersburg 199034, Russia; f_gol@mail.ru

**Keywords:** aberrant sex determination, chromosome painting, comparative cytogenetics, genome architecture, mandarin vole, microdissection, high-throughput sequencing, rearrangements, rodents, sex chromosomes

## Abstract

The mandarin vole, *Lasiopodomys mandarinus*, is one of the most intriguing species among mammals with non-XX/XY sex chromosome system. It combines polymorphism in diploid chromosome numbers, variation in the morphology of autosomes, heteromorphism of X chromosomes, and several sex chromosome systems the origin of which remains unexplained. Here we elucidate the sex determination system in *Lasiopodomys mandarinus vinogradovi* using extensive karyotyping, crossbreeding experiments, molecular cytogenetic methods, and single chromosome DNA sequencing. Among 205 karyotyped voles, one male and three female combinations of sex chromosomes were revealed. The chromosome segregation pattern and karyomorph-related reproductive performances suggested an aberrant sex determination with almost half of the females carrying neo-X/neo-Y combination. The comparative chromosome painting strongly supported this proposition and revealed the mandarin vole sex chromosome systems originated due to at least two *de novo* autosomal translocations onto the ancestral X chromosome. The polymorphism in autosome 2 was not related to sex chromosome variability and was proved to result from pericentric inversions. Sequencing of microdissection derived of sex chromosomes allowed the determination of the coordinates for syntenic regions but did not reveal any Y-specific sequences. Several possible sex determination mechanisms as well as interpopulation karyological differences are discussed.

## 1. Introduction

Most therian mammals have a conventional XX/XY sex chromosome system with the Y-borne testis-determining *SRY* gene. Nevertheless, several dozen species with nonstandard systems of chromosomal sex determination have been described among mammals [1]. There are species with isomorphic sex chromosomes in males and females (three species of *Ellobius* genus), with the absence of the regular Y chromosome (e.g., *Dicrostonyx torquatus*) or the *SRY* gene (e.g., *Ellobius lutescens, Tokudaia*), with the Y chromosome in females (e.g., *Myopus schisticolor*), with heteromorphism of the X chromosomes or multiple sex chromosomes (see more examples in [2]). Most species of mammals with aberrant sex chromosome systems belong to the subfamily Arvicolinae (Myomorpha, Rodentia). One such example is the mandarin vole, *Lasiopodomys mandarinus*.

The first karyotype descriptions of *L. mandarinus* made in the 1970s and further works showed the variability of chromosomal numbers among and within populations of this species. In the mandarin voles from Mongolia and Buryatia (*Lasiopodomys mandarinus vinogradovi*) the diploid chromosome number (2n) is 47–48 [3], whereas Chinese populations display 2n = 49–52 (*L. m. mandarinus*, Henan province [4,5,6]), 2n = 48–50 (*Lasiopodomys mandarinus mandarinus*, Shandong province [7]), or 2n = 47–50 (*Lasiopodomys mandarinus faeceus*, Jiangsu province [8]). Comparative cytogenetic studies made with G-banding and routine staining indicated intrapopulation variability in morphology of some chromosome pairs in karyotypes of *L. mandarinus,* specifically, two pairs of autosomes (No. 1 and No. 2) and sex chromosomes. Each of the studied populations is characterized by large heteromorphic X chromosomes that differ both in shape and size. Wang et al. [7] suggested that the unusual X chromosome variability in *L. mandarinus* originated through translocation of autosomes onto sex chromosomes. The autosomal polymorphism is subspecies-specific, not associated with sex and sex chromosomes, and caused by presumed inversions based on G-banding analysis [7,9]).

Sex chromosome systems of *L. m. vinogradovi* have been investigated first with G-banding and routine staining [3] and recently by cross-species chromosome painting [10]. Using the last method, Gladkikh et al. [10] demonstrated the origin of neo-X chromosomes by at least two independent autosome-sex chromosome translocation events. The complex of sex chromosomes in the only female (2n = 47) studied by these authors consisted of one metacentric chromosome (neo-X1), one submetacentric chromosome (neo-X2), and one small acrocentric (neo-X3). But at least two other sex chromosome systems exist in *L. m. vinogradovi*. In some females (2n = 47) the system is represented by the neo-X2 and two small acrocentrics. A male combination (2n = 48) is represented by the neo-X1 plus three small acrocentrics, one of which is considered to be the Y chromosome [3]. Both of these karyomorphs were described based on the examination of a relatively small sample with traditional methods unable to determine homology among small acrocentrics [3].

All studied males from the Chinese population had an unpaired acrocentric chromosome that could be a Y chromosome [7,8,11]. Analysis of the synaptonemal complex of *L. mandarinus* from China showed that there was indeed a chromosome that could pair with the X chromosome [12,13]. It was also shown that the sex chromosomes of the male *L. m. vinogradovi* pair and recombine at pachytene [14]. Studies on *L. m. mandarinus* demonstrated that sex determination in the subspecies is independent of *SRY* or *R-spondin 1* [13]. Chen et al. [15] also excluded the *Sall 4* gene as a potential testis-determining factor in this subspecies. Up to now, all attempts to find a chromosome carrying any Y chromosome-specific genes or regions (*SRY*, *Rbm*-gene family, PAR) in *L. mandarinus* using molecular approaches failed [5,15,16]. Thus, the question about the presence of a Y chromosome in karyotypes of male mandarin voles is actually controversial.

Despite the unusual heteromorphism of X chromosomes and failure to detect any testis-determining gene, the mandarin vole was, by default, considered as a species with a standard, XY males/non-Y females, sex determination system. Within the framework of this hypothesis, the absence of several predicted sex chromosome combinations (specifically, neo-X1/neo-X1 females expected in the progeny of males and females carrying neo-X1 chromosome, and neo-X2/Y males, expected in the progeny of males and females carrying neo-X2 chromosome) needs explanation. The failure to reveal these combinations may be either the consequence of small sample size or low viability of their carriers. In the latter case, the reduced fertility is expected for the neo-X2 females because three-quarters of their offspring from crossing with neo-X1/Y males (Y/0, neo-X1/0, and neo-X2/Y) should be nonviable. Also, under conventional sex determination with normal meiotic chromosome segregation, female carriers of a single neo-X2 should deliver only daughters. These predictions can be tested by the crossbreeding experiments.

To elucidate the sex determination system in *L. mandarinus vinogradovi,* we carried out a comprehensive study which combined several different conventional and molecular cytogenetic methods, single chromosome DNA sequencing, and breeding experiments revealing the chromosome segregation pattern as well as the reproductive performance of different karyomorphs. Comparative molecular cytogenetic research methods have been applied to achieve a more detailed description of the karyotype of this species and a deeper study of the autosomal polymorphism.

## 2. Materials and Methods

### 2.1. Ethics Statement

All applicable international, national, and/or institutional guidelines for the care and use of animals were followed. All experiments were approved by the Ethics Committee on Animal and Human Research of the Institute of Molecular and Cellular Biology, Siberian Branch of the Russian Academy of Sciences (IMCB, SB RAS), Russia (order No. 32 of 5 May 2017). This article does not contain any studies with human participants performed by any of the authors.

### 2.2. Specimens Sampled

In total, the karyotypes of 205 voles (163 females and 42 males) were examined with conventional cytogenetic methods. Of them, 27 individuals were captured in Selenginskii and Dzhidinskii districts of Buryatia in 2002–2017. The rest were captive-born descendants of these voles. Twelve animals (7 voles from the same laboratory colony and 5 voles captured in Selenginskii districts of Buryatia in 2017) were chosen for molecular cytogenetic study and chromosome sequencing.

### 2.3. Chromosome Preparation and Chromosome Staining

For karyotyping, chromosome suspensions were obtained from bone marrow and/or spleen by a standard method with preliminary colchicination of animals [17]. For some individuals, short-term culture of bone marrow was used. For molecular cytogenetic study, metaphase chromosome spreads were prepared from primary fibroblast cultures as described previously [18,19]. The fibroblast cell lines were derived from biopsies of skin, lung, and tail tissues in the case of laboratory animals and from finger biopsy in the case of wild animals as described previously [10]. All cell lines were deposited in the IMCB, SB RAS, cell bank (“The general collection of cell cultures”, No. 0310-2016-0002). Cell cultures and chromosome suspensions were obtained in the Laboratory of animal cytogenetics, the IMCB, SB RAS, Russia.

G-banding was performed on chromosomes of all animals prior to fluorescence in situ hybridization, using the standard trypsin/Giemsa treatment procedure [20]. C-banding has followed the classical method [21] or the method with some modifications [10,21].

### 2.4. Crossbreeding Experiments

We sexed 327 offspring delivered by 38 females and surviving to at least 24 days of age. The dams were karyotyped, and offspring sex ratios in the pooled progeny obtained from dams of each karyomorph were compared. For each female karyomorph, the observed ratio of male to female offspring in the pooled progeny was compared with an even sex ratio with Chi-square goodness-of-fit test or, in case of a small sample, with Fisher’s exact test. Female offspring (*n* = 64) born to 19 of the same dams were karyotyped, and the proportions of daughters holding different karyomorphs in the pooled progeny were calculated and compared between karyomorphs using 2 X 3 Fisher’s exact test.

### 2.5. Female Reproductive Success in Relation to Karyomorphs

Thirty-two virgin females older than 70 days were paired with unrelated unfamiliar males. All pairs were maintained under standard conditions (see [22] for details). The females were weighed weekly until the detection of pregnancy, after which the nests were checked every two days until delivery, and then again once a week. Thus, the litter sizes were determined no later than the second day after birth. The number of surviving offspring was determined at weaning (on Day 24 after birth). The pairs were monitored for three months. The dams were karyotyped immediately or within a few months after the end of this experiment. We estimated the effects of the dam’s karyomorph on the following characteristics of reproductive success over a three-month period: Proportion of females who gave birth (Fisher’s exact test), number of litters, total number of the delivered offspring, and total number of the weaned offspring; the last three parameters were determined and compared for those females who gave birth (Student’s *t*-test).

All tests were two-tailed and the α level of significance was 0.05.

### 2.6. Microdissection and Probe Amplification

G-banding by trypsin using Giemsa (GTG-banding) was performed before microdissection to accurately identify chromosomes. Glass needle-based microdissection was performed as described earlier [23]. One copy of each sex chromosome was collected. Chromosome-specific libraries were obtained using whole genome amplification (WGA) kits (Sigma). After amplification, DNA was purified using nucleic acid purification kits for DNA (BioSilica). DNA libraries were labeled using WGA. Chromosome-specific probes were obtained for chromosomes neo-X1 (probe L2), neo-X2 (probe L3) from LMAN19f, neo-X2 (probe L33) and neo-Y (probes L11 and L13 (distal part)) from LMAN14f, neo-X1 (probe L8) and neo-Y (probe L5) from LMAN15m, and neo-X2 (probe L31) from LMAN5f. Here, LMAN is an individual of *L. m. vinogradovi* (f, female; m, male).

### 2.7. Fluorescence in Situ Hybridization (FISH)

The sets of flow-sorted chromosomes of field vole (*Microtus agrestis,* MAG) and Arctic lemming (*Dicrostonyx torquatus,* DTO) painting probes were described previously [10,24,25,26,27]. The telomeric DNA probe was generated by PCR using the oligonucleotides (TTAGGG)_5_ and (CCCTAA)_5_ [28]. Clones of human ribosomal DNA (rDNA) containing partial 18S, full 5.8S, and a part of the 28S ribosomal genes and two internal transcribed spacers were obtained as described in Maden et al. [29]. FISH was performed following previously published protocols [30,31]. Images were captured using VideoTest-FISH software (Imicrotec) with a JenOptic charge-coupled device (CCD) camera mounted on an Olympus BX53 microscope. Hybridization signals were assigned to specific chromosome regions defined by G-banding patterns previously photographed and captured by the CCD camera. All images were processed using Corel Paint Shop Pro X3 (Jasc Software).

### 2.8. Sequencing

Libraries for sequencing were prepared according to the TruSeq Nano Library Preparation Kit (Illumina). Size selection was performed using the Pippin Prep. Quantification of the libraries before sequencing was performed using real-time PCR with SYBR GREEN. Then, 300-base pair paired-end reads were generated on Illumina MiSeq using the Illumina MiSeq Reagent Kit v3, according to the manufacturer’s instructions. Raw reads were deposited in the Sequence Read Archive of the National Center for Biotechnology Information under accession PRJNA613194.

### 2.9. Bioinformatic Analysis

The reads obtained by sequencing were used in the DOPseq_analyzer pipeline (https://github.com/ilyakichigin/DOPseq_analyzer) to search for syntenic regions in the mouse genome assembly GRCm38. The operation of this pipeline was reported by Makunin et al. [32]. It can be briefly described as follows. First, the cutadapt 1.18 tool [33] removes the sequences of Illumina adapters and primers used for amplification. The purified reads are aligned on the mouse genome GRCm38 (to search for target regions) and the human genome GRCh38 (to remove contamination reads) using Burrows-Wheeler Aligner 0.7.17 [34], low-quality alignments (alignment length <20, mapping quality <20) were discarded. Then, by calculation, the density of alignment and the identification of target regions occurs using DNAcopy package [35]. The resulting coordinates are then checked manually in the UCSC (University of California, Santa Cruz) genome browser (https://genome.ucsc.edu).

The obtained coordinates for syntenic blocks are slightly different between libraries since they contain different amounts and diversity of the target DNA. To establish more accurate averaged boundaries of the evolutionary breakpoints, the reads obtained for all libraries were combined and reused in DOPseq_analyzer.

## 3. Results

### 3.1. Sex Chromosome Combinations Revealed by Extensive Karyotyping

The karyotypes of the studied individuals included, in addition to 22 pairs of autosomes common to males and females, four combinations of large heteromorphic sex chromosomes and small acrocentric chromosomes unidentifiable with conventional cytogenetic methods.

All studied males (42 individuals) had 2n = 48 and an identical system of sex chromosomes: Neo-X1 and three small acrocentrics (karyomorph I, thereafter KI). Males with neo-X2 were not found. Among 163 females, three sex chromosome combinations were revealed. Karyomorph II (KII, 47% of the studied females) corresponded to the sex chromosome system described by Gladkikh et al. [10] and had 2n = 47 with neo-X1, neo-X2, and neo-X3. Karyomorph III (KIII, also 47% of the studied females) had 2n = 47 with neo-X2 and two small acrocentrics. Karyomorph IV (KIV, 6% of females) had 2n = 48 with neo-X1, neo-X1, and two small acrocentrics.

### 3.2. Hybridization Experiment

Males were presented in the pooled progeny of all female karyomorphs. Sex ratio was female-biased in offspring of KII and KIII females. To the contrary, only sons were born to the few breeding KIV females (Table 1). The differences in the offspring sex ratio between KIV and the other two karyomorphs were significant (Fisher’s exact test: KII vs. KIV, *p* = 0.001; KIII vs. KIV, *p* = 0.002). Sex ratio in the pooled sample, including the offspring of all females, was significantly female-biased (40% of males, χ^2^ = 12.96, df (degrees of freedom) = 1, *p* < 0.001).

Karyotyping of 19 dams and their 64 daughters showed that KII and KIII females produced mainly KIII and KII daughters, respectively. As one can expect, KIV females were not found among the daughters of KIII dams. This variant was very rare in the progeny of KII females. There was a significant difference between the two most common karyomorphs in the proportions of KII:KIII:KIV daughters in progeny (KII dams: 9:25:3; KIII dams: 21:6:0; *p* < 0.001).

### 3.3. Female Reproductive Success Related to Their Karyomorphs

Of the 32 females participating in the experiment, 13 (41%) belonged to KII, 15 (47%) to KIII, and four (13%) to KIV. Female carriers of the two most common karyomorphs did not differ in any measure of reproductive success (Appendix A). At the same time, none of the rare KIV females produced offspring during a three-month period. This karyomorph significantly differed from the other two in the proportion of carriers that gave birth (Fisher’s exact test for KII vs. KIV: *p* = 0.002; KIII vs. KIV: *p* = 0.009) (Appendix A).

### 3.4. Comparative Molecular Cytogenetic Investigation of Different L. m. vinogradovi Karyomorphs

Comparative chromosome painting with two sets of painting probes was used for the analysis of karyotypes of 12 animals (Table 2). Since the set of *M. agrestis* probes showed almost complete identity of the autosomal sets in various individuals of *L. m. vinogradovi*, only partial localization of the *D. torquatus* probes was carried out on the chromosomes of most individuals. Application of comparative chromosome painting allowed us to establish that the acrocentric chromosomes participating in formation of complex sex chromosome systems in *L. m. vinogradovi* are homologous to MAG13/X/13 (designated here as neo-Y) and MAG17/19 (designated here as neo-X3, according to [10]) (Figure 1 and Figure 2).

Sex chromosomes of KI (males) were represented by the largest metacentric (neo-X1) and a small-sized acrocentric (putative neo-Y) (Figure 1a). The autosomes homologous to MAG17/19 (neo-X3) should also be included in the complex of male sex chromosomes as they were present in single copy in KII. The MAGX probe hybridized to the p-arm of the neo-X1 chromosome and to the interstitial part of neo-Y chromosome (Figure 2e). MAG13 labeled the q-arms of neo-X and neo-Y (Figure 1a). The neo-X1 chromosome had three C-positive blocks on the q-arm (Figure 3a,b).

Sex chromosomes of KIII (females) were represented by the large neo-X2, one small unpaired acrocentric corresponding to neo-X3, and another small unpaired acrocentric homologous to MAG13/X/13 (neo-Y) (Figure 1c). As in the case of KII the neo-X2 chromosome had a block of grey heterochromatin in the area homologous to MAGX (Figure 3b,c)

KIV (females) have not been found among the animals analyzed by molecular cytogenetic methods but the structure of their karyotype can be unequivocally reconstructed based on the analysis of other karyomorphs. Their sex chromosome complex can be described as a pair of neo-X1 chromosomes and two acrocentrics homologous to MAG17/19 (neo-X3).

MAGY probe labeled heterochromatic, C-positive parts of neo-X1 and neo-X2 chromosomes of male and female *L. m. vinogradovi*.

The field vole and the Arctic lemming painting probes showed the conservatism of autosomal sets in the species and the same localization of the probes as was shown previously for LMAN0f with the only exception the chromosome 2 [10]. The chromosome is designated here as LMAN2 without an additional letter. We described four variants of the distribution of syntenic blocks homologous to MAG1 and MAG5 onto LMAN2 due to inversions: Both homologs are acrocentrics with the order of probes MAG1/5 (LMAN6f, 10m, 16f) or MAG1/5/1/5/1/5 (LMAN0f, 1f), or MAG1/5 and MAG1/5/1/5/1/5 (LMAN2f, 3f, 5f). Both homologs are submetacentric with the order of probes MAG5/1/5/1 (LMAN19f). The LMAN2 is heteromorphic, and this heteromorphism is formed by an acrocentric chromosome homologous to MAG1/5 and a submetacentric chromosome with a derived structure MAG5/1/5/1 (LMAN14f, 15m, 17f) (Figure 1 and Figure 2, Table 2).

Noncentromeric interstitial telomeric sequences (ITS) were localized on the autosome 1 and on the neo-X2 chromosome of all studied specimens. Each individual carried three rDNA clusters located on chromosomes 2, 18, and 22 (Figure 2). The size of rDNA clusters was different on homologs of autosome 2.

In general, no correlation was observed between sex chromosome systems and the morphology of chromosome pair 2, both among wild-caught and captive-born animals (Table 2).

### 3.5. Sequencing and Bioinformatic Analysis

Eight sex chromosome libraries were sequenced on the Miseq Illumina platform (Appendix A). All sequencing and alignment statistics are presented in Appendix A. Based on the low-coverage sequencing data obtained, the coordinates of large syntenic regions and the boundaries of evolutionary rearrangements relative to the mouse (*Mus musculus*, MMU) genome were identified (Appendix A).

False-positive regions were detected on the chromosome neo-Y where the painting probe of MAGX chromosome labeled a small interstitial region (Figure 2a,c). The sequencing data did not show homology of the chromosome to the MMUX but indicated the synteny of the neo-Y chromosome with a region of the MMU18 only.

## 4. Discussion

The fascinating sex chromosome polymorphism of the mandarin vole raises three closely related questions: (1) How is such an unusual system maintained in a population? (2) How did the system evolve, that is, what chromosome rearrangements led to the polymorphism of sex chromosomes? (3) What is the specific (molecular) mechanism of sex determination in this species?

Up to now, the investigators mainly addressed the latter two problems. Here, on the contrary, we found it useful to focus on the first two questions, and our results provided the premise to approach solving the third, and perhaps, the main question for many evolutionary biologists.

### 4.1. How Is Such an Unusual System Maintained in Populations of L. m. vinogradovi?

If one assumes the presence of a single neo-Y chromosome to be both necessary and sufficient to initiate testis development, the described variation of sex chromosome combination is hard to explain. Why some of the expected karyomorphic variants are missing, and where do the females with a single X chromosome (KIII) come from?

Theoretically, the offspring resulting from a cross between the females with the neo-X1 chromosome and males also carrying the neo-X1 chromosome should include neo-X1/neo-X1 females. These females have not been revealed in previous studies, but here we did find this rather infrequent karyomorph (KIV) due to comprehensive sampling. At the same time, no males with neo-X2/neo-Y, which are expected to occur among both KII and KIII females’ progeny, were detected among more than 40 karyotyped male voles. From the “standard sex determination” point of view, their absence may be explained by the lethality of male embryos with neo-X2/neo-Y combination. However, this means that the KIII females produce three-quarters of nonviable embryos (neo-Y/0, neo-X1/0, and neo-X2/neo-Y), and KII females produce one-quarter of the nonviable embryos (neo-X2/neo-Y). According to this hypothesis, only KIV females are lucky to have no costs associated with nonviable offspring and, therefore, are predicted to have the highest reproductive success. This prediction does not seem to be supported by the results of our experiments. Further and most important, under the standard sex determination and normal sex chromosome segregation, KIII dams should produce only KII daughters. In fact, they also produce KIII daughters and KI sons. Finally, the finding that KII dams delivered a large proportion of KIII daughters also needs explanation. Within the framework of the standard sex determination hypothesis, the observed patterns require both a nondisjunction of Y chromosome in the second division and nonviability of most karyomorphs. This scenario appears to be very unlikely. On the other hand, the chromosome segregation pattern inferred from our results appears to be in good agreement with the “XY female hypothesis” [36,37,38]. This hypothesis requires neither chromosome nondisjunction nor the lethality of large proportions of offspring; the only one nonviable combination (neo-Y/neo-Y) is implied. The unequal proportions of different karyomorphs in progeny of each type of females may be explained by two phenomena. The first one is apparent lower viability of neo-X1/neo-X1 females due to, for example, a violation in gene dosage compensation, or some other unknown causes. Unfortunately, the mechanisms of gene dosage compensation in this species are unexplored. The second plausible phenomenon is that the relative success of male neo-X1 and neo-Y spermatozoa depends on the karyomorph of a fertilized female. From our chromosome segregation data, it looks like male gametes carrying neo-Y are favored in neo-X1/neo-X2 females, resulting in higher than expected proportions of both sons (neo-X1/neo-Y) and neo-X2/neo-Y daughters (each about 40% vs. the expected 25%). To the contrast, an excess of sons (40% vs. 33% expected) and neo-X1/neo-X2 females (40% vs. 33% expected) in a progeny of neo-X2/neo-Y dams and predominance of sons in a progeny of neo-X1/neo-X1 (100% in our small sample) dams suggest the spermatozoa with neo-X1 chromosome to be favored in these crossbreeding combinations. The cytogenetic mechanisms underlying these phenomena are unclear but it is noteworthy that they all reduce a sex ratio distortion. Thus, in terms of ultimate causes, these mechanisms might be selected as they increased the investment to sons which have higher reproductive value than daughters in a population with a female-biased primary sex ratio [39].

Our proposition that neo-X2/neo-Y mandarin voles are actually present but have a female phenotype received strong support from the results of fluorescence in situ hybridization and the comparative molecular sex chromosome investigation.

### 4.2. Complex Systems of Sex Chromosomes in L. m. vinogradovi and Their Origin

The comparative chromosome painting convincingly showed that at least two autosomal translocations on sex chromosomes took place in the evolution of the mandarin vole karyotypes forming the neo-X and neo-Y chromosomes in *L. m. vinogradovi*.

Comparing our sex chromosome sequencing data to previous comparative chromosome painting data [26] confirms that MAG13=MMU18, MAG17=MMU13/15, and MAG19=MMU15. However, our neo-Y sequencing data did not show homology of this chromosome to the mouse X chromosome, and this contrasts to the clear detection of X chromosome signal on the neo-Y chromosome by FISH. Therefore, it is likely that the FISH-signal represents shared repetitive sequences that are not included in the bioinformatic analyses. The fact that an unpaired chromosome with a small interstitial block of heterochromatin was detected by C-banding in karyotypes of both males and females further confirmed the presence of a block of repeated sequences on the chromosome (Figure 3a,c).

The regular Y chromosome has not been revealed in comparative chromosome painting experiments based on localization of the *M. agrestis* Y chromosome probe. Previously Zhao et al. [40] also failed to find a regular Y by FISH experiments with localization of partial human and whole mouse Y. Detection of signals from the MAGY probe in the heterochromatic, C-positive, parts of neo-X1 and neo-X2 chromosome of *L. m. vinogradovi* males and females may be caused by repeated sequences. As the Y chromosome of *M. agrestis* carries a huge block of heterochromatin, it can be assumed that both these arvicoline species have similar repeated sequences on their sex chromosomes. This assumption does not exclude the presence of sequences responsible for masculinization function in this area. This phenomenon requires a thorough study.

By analyzing the low-coverage chromosome sequencing data, we also failed to identify any Y chromosome-specific genes or regions. This may indicate either the elimination of the regular Y chromosome in this species or the insufficiency of our approach of aligning the reads of chromosomes on the mouse genome to search for the Y chromosome due to its rapid evolution and complex repetitive structure. To date, there are no Y chromosome assemblies for the representatives of Cricetidae family; it is possible that the forthcoming release of such genomic assemblies will allow us to answer the question about the presence of a regular Y chromosome using bioinformatic comparative genomic analysis methods.

It should be noted here that *L. mandarinus* is one of the most unusual species in terms of the synaptic behavior of its sex chromosomes. The nature of the XY pairing observed in this species differs markedly from that revealed in all other arvicolines. It was proposed based on the recombination pattern detected in pachytene that the XY synapsis in *L. m. vinogradovi* is a derivative condition resulting from de novo translocated autosomal material [14]. The FISH results obtained here completely confirm the suggestion. Preservation of ITS at the confluence sites of ancestral autosomes and sex chromosomes also indicates that the translocation has occurred recently.

### 4.3. Possible Mechanisms of Sex Determination in L. m. vinogradovi

The pattern of association between phenotypic sex and sex chromosome combinations found in *L. m. vinogradovi* is similar to that in the wood lemming (*Myopus schisticolor*), collared lemmings (genus *Dicrostonyx)*, and the African pygmy mouse (*Mus minutoides*) [41,42,43,44]. It suggests an X-linked mutation (in *L. m. vinogradovi*, the mutation on the neo-X2 chromosome) to prevent masculinization of neo-X2/neo-Y individuals, but exact genetic bases of the male-to-female sex reversal are unknown. In the case of the mandarin vole, the problem of sex determination mechanism is additionally complicated by the failure to reveal any Y-specific genes or regions. The following scenarios can be suggested.

(1)The neo-X2 chromosome contains an epistatic locus (B) suppressing the dominant male development trigger (A) located on the neo-Y (Figure 4a).(2)The neo-X1 chromosome contains a locus (D) complementing the male development trigger (C), whereas this locus is absent from or is inactive on the homologous neo-X2 (Figure 4b). In this case, the sex-determination system is similar to that described for *Myopus schisticolor* [41] where a deletion differentiates two types of X chromosomes [43]; the same has been suggested for *Dicrostonyx torquatus* [42]. Zhu et al. [8] proposed a possible role of deletions in the formation of sex chromosomes in *L. m. faeceus*.(3)Taking into account the uncertainty of the location of male development gene(s), it cannot be ruled out that they are associated not with neo-Y but with neo-X1 chromosome only (Figure 4c). According to this scenario, the neo-X2 chromosome either is capable of inactivating neo-X1 or carries a trigger-suppressing gene resulting in female phenotype of neo-X1/neo-X2 carriers Although nonrandom inactivation of the X chromosome has not been described for the mandarin voles, it has been identified in experiments on interspecific crosses of several arvicoline species [45]. This scenario, however, appears to be the least plausible as it requires the dominant male development trigger to be somehow inactive in double doses to produce neo-X1/neo-X1 females.

Whichever of the proposed scenarios is true, we believe that it should be the same in *L. m. vinogradovi* and *L. m. faeceus*. This assumption is based on the fact that the studied sample of voles from this Chinese population (Henan province) was represented by the same combinations of large X chromosomes and small acrocentrics, and in approximately the same proportions. In this population, a male karyomorph, KI, and female karyomorphs, KII and KIII, were common while KIV females were found as a rare variant [5]. In our opinion, information on the karyotypes of the mandarin voles from another Chinese population, Shandong province, deserves special attention. According to Wang et al. [7], females with a single X chromosome (KIII), common in other populations, were not found here, whereas an additional male karyomorph corresponding KIII has been reported. Thus, we assume that the aberrant sex determination system, in which some of the carriers of the Y chromosome display a female phenotype, has either not yet appeared or has already disappeared in this population. In our opinion, a detailed study of the sex chromosomes of voles from Shandong could shed light on the evolution of molecular mechanisms of sexual determination in this species.

### 4.4. Chromosomal Differences of L. mandarinus from Different Populations

Comparison of *L. mandarinus* showed clear differences in the karyotypes of individuals from different populations. So, the mandarin voles from Mongolia and Buryatia (*L. m. vinogradovi*), having diploid chromosome number 2n = 47–48, carried three pairs of metacentric chromosomes corresponding to pairs 1, 4, and 18 described in this work ([3], present data). The sizes of two types of chromosomes bearing regions homologous to MAGX, were approximately the same. All animals analyzed in the current work had stable and identical pairs of chromosome 1 (LMAN1), presented by two metacentric chromosomes. Both homologs carried interstitial telomeric sequences (ITS) in q-arms, separating synteny MAG8/21 and MAG11/13 (Figure 2). As previously suggested, that the association MAG11/13 is ancestral for the subgenus *Lasiopodomys*, the presence of ITS showed that the first pair of metacentric chromosomes was formed by the evolutionary recent fusion of the two ancestral pairs of chromosomes [10].

LMAN2 is homologous to MAG1 and MAG5. This fusion is characteristic of *L. m. vinogradovi* and it has never been found in any other arvicolines. The pair was polymorphic in the animal analyzed here due to multiple para- and pericentric inversions. It is important to notice here that both homologs of LMAN2 carry big clusters of ribosomal genes in the distal part of q-arms. It is possible that the fusion MAG1/5 is also characteristic of mandarin voles from other populations, but molecular cytogenetic methods must be used to verify this assumption.

In the karyotypes of all the studied individuals from China, there are only two stable pairs of bi-armed autosomes (corresponding to LMAN1 and 4 described in this work). Polymorphism of LMAN1 revealed in the Chinese population is not characteristic for individuals from the Buryatia. Zhang and Zhu [46] postulated that a Robertsonian fission is the main reason for the polymorphism of chromosome 1.

*L. m. faeceus* inhabiting the Jiangsu province in China has 2n = 47–50. The pair of largest autosomes is formed by two submetacentrics [8]. In the absence of molecular cytogenetic studies, it is difficult to state unequivocally, but it seems that sex chromosome systems are similar to those described in this work. Nevertheless, the relative sizes and ratio of lengths of arms of the submeta- and metacentric chromosomes attributed to the sex chromosomes are different, which may indicate a different accumulation of repeated sequences, or the presence of intrachromosomal rearrangements. It cannot be ruled out that other pairs of autosomes participated in the translocation of X chromosomes to autosomes. However, there is a polymorphism in one pair of autosomes in *L. m. faeceus*, apparently smaller than a pair of chromosomes 2 in *L. m. vinogradovi*. So, the autosomal polymorphism, previously described only for pairs of chromosomes 1 and 2, possibly affects other pairs of autosomes in karyotypes of the mandarin voles from different populations.

It is shown that *L. m. mandarinus* from Henan province (China) has a diploid number 2n = 49–52 [4,5,6], while the same subspecies in the Shandong province (China) has 2n = 48–50 [7]. Based on G-banding comparison we propose that the pair of chromosomes 1 in [7] is homologous to our LMAN1q. We should also note that the number of chromosomes bearing nucleolus organizer regions identified by Wang et al. [7] and in this work is different (4 vs. 3). Moreover, clusters of ribosomal genes are located on a pair of chromosomes 1 [7] (which corresponds to localization on LMAN1q). The morphology of sex chromosomes in *L. m. mandarinus* is similar to that described for *L. m. faeceus*. Surprisingly, among *L. m. mandarinus*, some males with a sex chromosome system morphologically similar to the karyomorphs III (females) described in this work were found [7], whereas among *L. m. faeceus*, females were discovered whose sex chromosome system was represented by two large submetacentric chromosomes [5].

It is known that the number of rDNA clusters and their localization can vary on various chromosomes even between closely related species [47]. This instability can be caused by a clustered structure of ribosomal genes that facilitate translocations by illegitimate recombination between nonhomologous chromosomes. Among mammals, multiple cases of interspecific variation in localization of rDNA clusters were described including presence on sex chromosomes [48]. Two of the three pairs of chromosomes carrying clusters of ribosomal genes in *L. m. vinogradovi* had stable morphology and localization of probes. At the moment, it remains unclear whether the polymorphism of the pair of autosomes 2 in *L. m. vinogradovi* and pair of autosomes 1 in *L. m. mandarinus* is associated with the location of the cluster of rDNA on them. The reasons for the significant polymorphism of these particular pairs of autosomes remain unclear.

In addition to the differences described above, individuals from different populations exhibit a different amount and distribution of heterochromatin ([5,7]; this work).

Thus, the karyotypes of the mandarin vole from all currently studied geographical populations are significantly different. In order to give a taxonomic assessment of these differences, it is necessary to study the karyotypes of *L. mandarinus* from different populations by molecular cytogenetic methods as well as applying molecular genetic data for the establishment of phylogenetic relationships between populations.

## 5. Conclusions

Euchromatic parts of mammalian sex chromosomes are highly conserved. Only rare cases of their involvement in rearrangements have been described in myomorph rodents, bats, carnivores, primates, and cetartiodactyls. Mandarin voles undoubtedly represent a unique species even among myomorphs and arvicolines. Their karyotypic features, such as the presence of different polymorphic pairs of autosomes and nonstandard sex chromosome systems, indicate significant plasticity of their genome, as well as ongoing processes of karyotypic evolution within the species. Such a diversity of sex chromosome systems as found in mandarin voles (within the same species) seems unique and has not been described yet in any other mammalian species. Such factors as modifications of the epigenetic state of DNA and accumulation of a large number of repeats may be required to trigger evolutionary plasticity [49].

## Figures and Tables

**Figure 1 genes-11-00374-f001:**
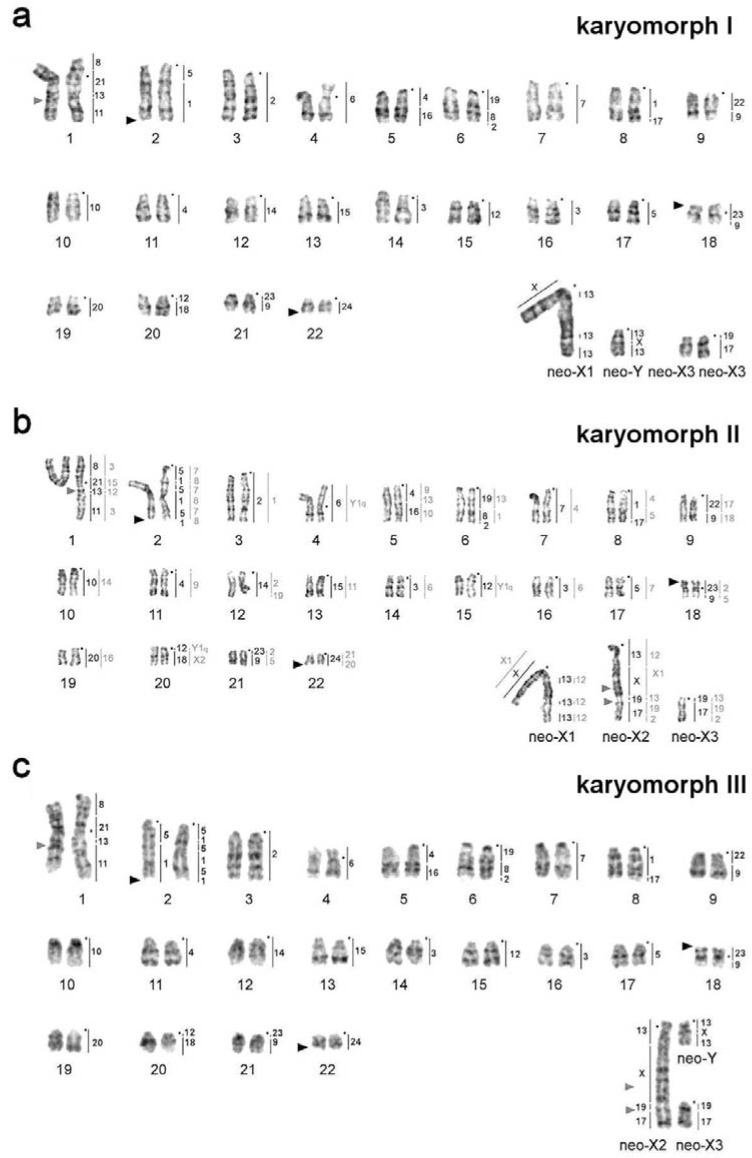
GTG-banded karyotype of *L. m. vinogradovi*: (**a**) LMAN10m, (**b**) LMAN0f [10], (**c**) LMAN5f. Black dots mark the position of centromeres. Vertical black bars mark the localization of *Microtus agrestis* (MAG) chromosome painting probes, vertical grey bars mark the localization of *Dicrostonyx torquatus* (DTO) painting probes. Numbers along the vertical lines correspond to chromosome numbers of *M. agrestis* and *D. torquatus*. Black triangles indicate sites of localization of ribosomal DNA clusters. Grey triangles indicate localization of the largest interstitial telomeric block.

**Figure 2 genes-11-00374-f002:**
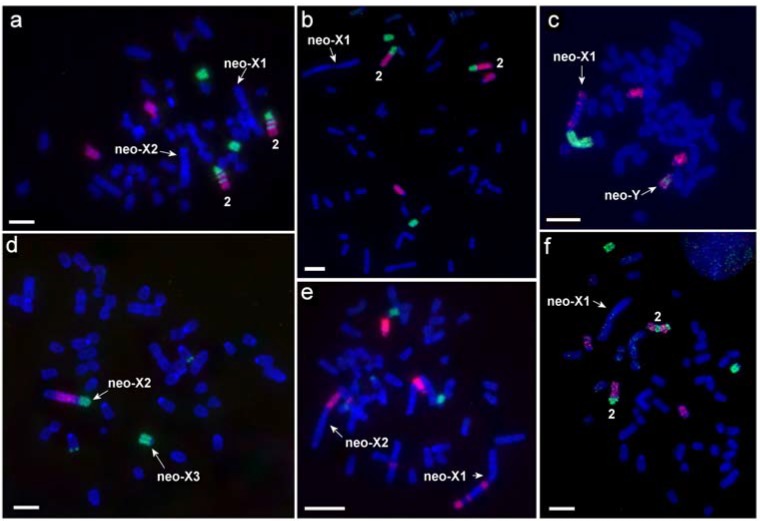
Examples of fluorescent in situ hybridization: (**a**) MAG5 (green) and MAG1 (red) onto LMAN1f, (**b**) MAG5 (green) and MAG1 (red) onto LMAN10m, (**c**) MAGX (green) and MAG13+14 (red) onto LMAN10m, (**d**) MAG17 (green) and MAGX (red) onto LMAN6f, (**e**) MAG23 (green) and MAG13+14 (red) onto LMAN1f, (**f**) MAG1 (red) and MAG5 (green) onto LMAN15m. Scale bar is 10 µm.

**Figure 3 genes-11-00374-f003:**
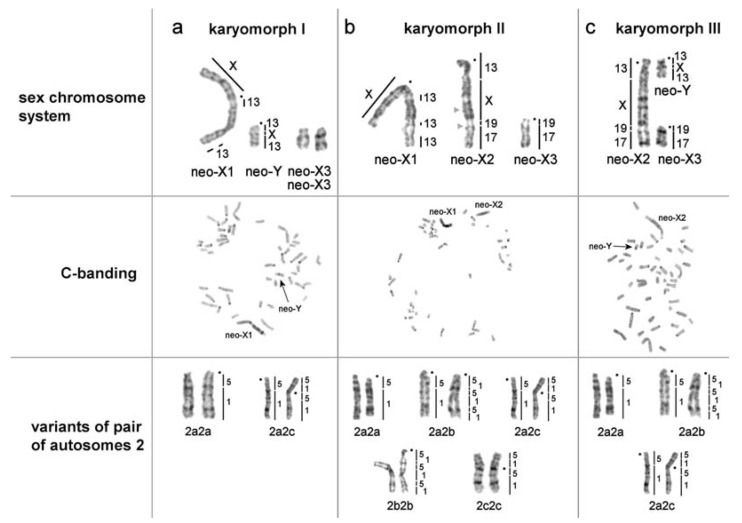
Polymorphic sex chromosomes and a pair of autosomes 2 in karyotypes of *Lasiopodomys mandarinus vinogradovi*: (a) Males (karyomorph I), (b) females, karyomorph II, (c) females, karyomorph III. From top to bottom: Sex chromosome system in the karyomorph, C-banding of metaphase chromosomes (chromosomes of LMAN10m (a), LMAN19f (b), and LMAN14f (c) are given as an example for each karyomorph), a pair of autosomes 2 of different individuals with localization of *Microtus agrestis* samples. Black arrows marked possible neo-Y.

**Figure 4 genes-11-00374-f004:**
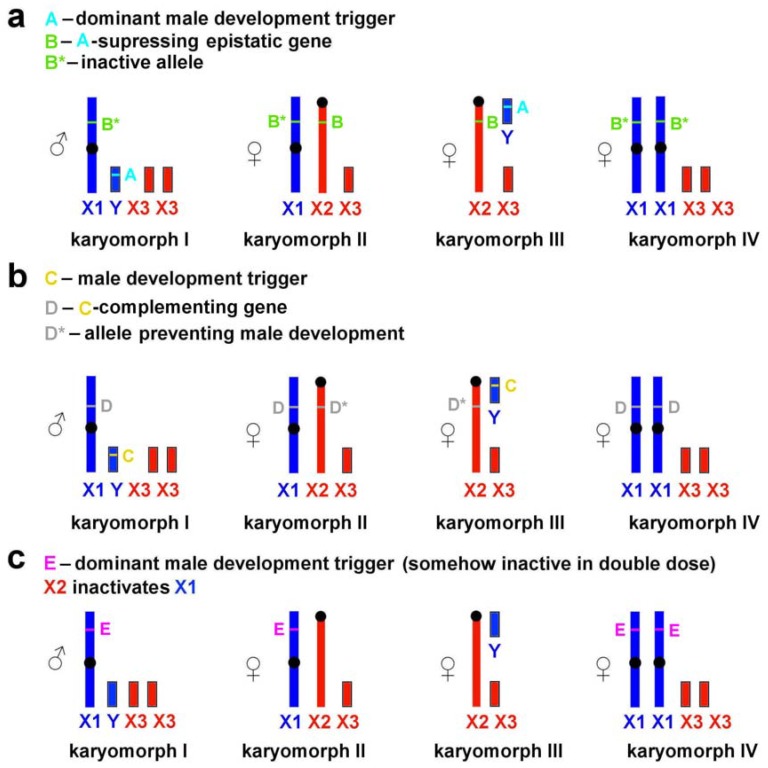
Possible mechanisms of sex determination in *Lasiopodomys mandarinus vinogradovi*: (**a**) a mechanism suggesting the presence of an epistatic locus on neo-X2 (X2) chromosome suppressing the dominant male development trigger on neo-Y (Y) chromosome, (**b**) a mechanism suggesting the presence of a locus on neo-X1 (X1) chromosome complementing the dominant male development trigger on neo-Y, (**c**) a mechanism suggesting the presence of male development trigger on neo-X1 chromosome and nonrandom inactivation of neo-X chromosomes. See comments in the text.

**Table 1 genes-11-00374-t001:** Comparison of sex ratio in the progeny born to the female carriers of different karyomorphs

Dam’s Karyomorph (number of dams)	Number of Sons/Number of Progeny (%)	Deviation from the Expected = 0.5
II (17)	54/149 (36.2)	χ^2^ = 11.2; *p* < 0.001
III (18)	70/171 (40.9)	χ^2^ = 5.66; *p* = 0.017
IV (3)	7/7 (100)	Fisher’s exact test: *p* = 0.070

**Table 2 genes-11-00374-t002:** The list of investigated individuals of *Lasiopodomys mandarinus vinogradovi* with abbreviated names, diploid numbers (2n), origin, systems of sex chromosomes, and types of autosome LMAN2; f, female; m, male; 2a, the acrocentric with the order of syntenic blocks MAG1/5; 2b, the acrocentric with the order of syntenic blocks MAG1/5/1/5/1/5; 2c, the submetacentric with the order of syntenic blocks MAG5/1/5/1. See comments in the text.

Abbreviation	2n	Origin	Complex of Sex Chromosomes	Type of Autosome LMAN2
LMAN0f	47	laboratory colony	neo-X1/neo-X2/neo-X3	2b2b
LMAN1f	47	laboratory colony	neo-X1/neo-X2/neo-X3	2b2b
LMAN2f	47	laboratory colony	neo-X1/neo-X2/neo-X3	2a2b
LMAN3f	47	laboratory colony	neo-X1/neo-X2/neo-X3	2a2b
LMAN5f	47	laboratory colony	neo-X2/neo-X3/neo-Y	2a2b
LMAN6f	47	laboratory colony	neo-X2/neo-X3/neo-Y4	2a2a
LMAN10m	48	laboratory colony	neo-X1/neo-Y/neo-X3/neo-X3	2a2a
LMAN14f	47	wild	neo-X2/neo-X3/neo-Y	2a2c
LMAN15m	48	wild	neo-X1/neo-Y/neo-X3/neo-X3	2a2c
LMAN16f	47	wild	neo-X1/neo-X2/neo-X3	2a2a
LMAN17f	47	wild	neo-X1/neo-X2/neo-X3	2a2c
LMAN19f	47	wild	neo-X1/neo-X2/neo-X3	2c2c

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
