# Peer review of "Complex Structure of Lasiopodomys mandarinus vinogradovi Sex Chromosomes, Sex Determination, and Intraspecific Autosomal Polymorphism"

_genes, 2020, doi:10.3390/genes11040374_

Round 1

Reviewer 1 Report

Romanenko et al provide a very detailed analyses of the karyotype variability, sex chromosome inheritance patterns, and sex chromosome composition in the mandarin vole. They have performed breeding experiments, extensive karyotyping, FISH analyses and sex chromosome sequencing analyses. The results shed light on how the different karyotypes are linked to sex, which type of inheritance patterns of sex chromosomes occur and with what frequency, and provide some first insight into the structure/composition of the (neo)sex chromosomes. This allowes them to come up with interesting hypothesis regarding the possible sex determination system in this interesting species.

Overall I think the data are convincing and relevant. However I have a few major comments:

1) Overall the paper is difficult to read. I understand that the complexity of the karyotypes, including variability in small autosomes being present, makes it challenging to describe matters in a very straithforward manner but I think there is definitely room for improvement.

I suggest that after the initial introduction of the different mandarin vole karyotypes (up to line 63 I would already indicate that this paper focuses on L. m. vinogradovi and start describing the karyotypes from there onwards using only a single type of nomenclature for the chromosomes. I propose they explain this work is a follow up of a previous analysis (Gladkikh et al., ref 10) where three different neo X chromosomes, and a putative neo Y chromosome were identified. Then which combinations have so far been described, which would also explain the total number of chromosomes better (now this is for example confusing: “«XsXm» (female, 2n=47), «XsO» (female, 2n=47)” since there is only one X but still 47 chromosomes, so if the ,  that

Then it could be explained that although existence of a neo-Y is proposed based on the observed karyotype(s), the actual presence of a true Y is still controversial (lines 84-93).

Lines 98-101 of the introduction are also rather confusing/complicated and could be described in a more comprehensive way. These are just suggestions, not mandatory changes, other adaptations might be chosen, as long as the paper becomes more readable

2) If the results start with the results described in 3.4, All the a/a/a confusion can perhaps be avoided? However, these results need more precise description. “molecular cytogenetic methods” is too vague, and it is not even fully clear from the text that all the data in Fig 1 are new. Please explain how probes from different species were used to determine syntenic regions, MAG13/X/13 is not explained in the main text (not even the abbreviation MAG), and in Fig2 it would be very helpful to indicate the signals corresponding to the chromosome 2, and each of the different sex chromosomes (X1, X2, X3, neoY1). Also please explain why telomeric and ribosomal DNA probes were used.

3) the sequencing of dissected probes is a very nice addition but not clearly described. The authors refer to a methods chapter, but it would still be necessarily to give a very brief description of the material used, the dissection method (how were the chromosomes recognized), and if there was any further purification before proceeding with the WGA. In addition, I presume that the limited depth of sequencing (coverage was not reported) precludes identification of genes? Please explain why there is no information on numbers of genes found, and on whether known critical genes involved in sex determination are present etc? Will the sequencing data be deposited and accessible to other researchers? I would also appreciate more explanation of the implications of the sequencing results. For example, the neoY appears to cover a region syntenic to mouse chromosome 18. This region is also present on X1 and X2, so if there is a dominant sex determining gene, it should be localized in this region. Perhaps this can be explained.

Minor comments:

General: your sequencing data nicely show that almost the complete mouse X chromosome is still present on neoX1 and X2. However, these chromosomes also now include “autosomal” parts. What is known about dosage compensation mechanisms in these animals, is only part of the X heterochromatic/barr body-like in females? Is X1 preferentially inactivated and could the relative infrequent occurrence of the type IV karyotype be related to lethality associated with inactivating both X chromosomes? The authors do not need to go into this, but I just wondered what is known about this and whether their data shed any light on dosage compensation mechanisms in these animals.

Lines 244-246:” The autosomes homologous to MAG 17/19 (neo-X3) should also be included in the complex of male sex chromosomes as their homologs were found unpaired in KII.”

The sentence is confusing and the term unpaired is confusing, do the authors mean that the neoX3 is present in single copy?

Fig 1: Please indicate KI, KII and KIII in the figure with the appropriate karyotype. Also, why are results from DTO probes only shown in b? Also please clearly refer to the figure 2b where DTO probe results are mentioned in the main text?

Fig 3 legend is unclear “females of one type” females of another type” Please indicate which type. The phrasing also suggests that results from only 3 animals are shown, whereas multiple pairs of chromosome 2 are shown in each of the three columns. So please explain more clearly what is shown.

Lines 276-278:” acrocentrics homologous to MAG17/19 (neo-X3). MAGY probe labeled heterochromatic, C-positive parts of neo-X1 and neo-X2 chromosomes of male and female L. m. vinogradovi”

Where is this result shown? There is no Y indicated in the lines representing the MAG chromosomes in Fig 1?

The list of probes in TableS2 does not correspond to the probes listed in table S3. In addition, it is not clear why samples are labelled “double” L9/L9, in addition, the significance of adding “new” is not clear. The Page name is in Russian. Please be more precise.

Line 300-301: this is not very clearly phrased. The sequencing results indicate synteny of the neo-Y chromosome with a region of mouse chromosome 18.

Lines 347-352: “The sequencing data confirmed the comparative chromosome painting data showing that MAG13=MMU18, MAG17=MMU13/15, MAG19=MMU15 [26] with one exception. The false-positive was detected by FISH on the neo-Y chromosome. As the sequencing data did not show homology of this chromosome to the mouse X chromosome, we supposed that the detection of the region was caused by FISH-based localization of shared repetitive sequences, while applied here the bioinformatic 351 analysis involved only unique, unrepeated sequence.”

This is unclear, perhaps rephrase along these lines:

Comparing our sex chromosome sequencing data to previous comparative chromosome painting data [26] confirms that MAG13=MMU18, MAG17=MMU13/15, and MAG19=MMU15. However, our neoY sequencing data did not show homology of this chromosome to the mouse X chromosome, and this contrast to the clear detection of X chromosome signal on the neo-Y chromosome by FISH. Therefore, it is likely that the FISH-signal represents shared repetitive sequences that are not included in the bioinformatic analyses.

Author Response

Since the individual comments of the first reviewer included many questions, we divided the answers into separate parts to give an answer to each question.

1)Overall the paper is difficult to read. I understand that the complexity of the karyotypes, including variability in small autosomes being present, makes it challenging to describe matters in a very straithforward manner but I think there is definitely room for improvement.

I suggest that after the initial introduction of the different mandarin vole karyotypes (up to line 63 I would already indicate that this paper focuses on L. m. vinogradovi and start describing the karyotypes from there onwards using only a single type of nomenclature for the chromosomes. I propose they explain this work is a follow up of a previous analysis (Gladkikh et al., ref 10) where three different neo X chromosomes, and a putative neo Y chromosome were identified. Then which combinations have so far been described, which would also explain the total number of chromosomes better (now this is for example confusing: “«XsXm» (female, 2n=47), «XsO» (female, 2n=47)” since there is only one X but still 47 chromosomes, so if the , that

Then it could be explained that although existence of a neo-Y is proposed based on the observed karyotype(s), the actual presence of a true Y is still controversial (lines 84-93).

Lines 98-101 of the introduction are also rather confusing/complicated and could be described in a more comprehensive way. These are just suggestions, not mandatory changes, other adaptations might be chosen, as long as the paper becomes more readable

The introduction was partly rewritten (lines 54-99). We hope it became more readable.

2) If the results start with the results described in 3.4, All the a/a/a confusion can perhaps be avoided?

We changed text to avoid the designations.

However, these results need more precise description. “molecular cytogenetic methods” is too vague,

We added refinement in the text (line 232), indicated that the method of comparative chromosome painting was used.

and it is not even fully clear from the text that all the data in Fig 1 are new.

The karyotype of  LMAN0f presented here in the Figure 1b was published in our previous work (Gladkikh et al. 2016). The reference was added in the capture to the Figure 1 (line 253).

Please explain how probes from different species were used to determine syntenic regions,

A set of Microtus agrestis painting probes is widely used to compare karyotypes of different arvicoline species. However, the use of additional sets of probes makes it possible in some cases to identify additional, finer rearrangements within conservative segments.

MAG13/X/13 is not explained in the main text (not even the abbreviation MAG),

The corresponding abbreviations and their interpretation have been added to section 2.7. (line 156).

and in Fig2 it would be very helpful to indicate the signals corresponding to the chromosome 2, and each of the different sex chromosomes (X1, X2, X3, neoY1).

In the Figure 2 we marked the chromosomes 2, X1, X2, X3, and Y where they can be identified.

Also please explain why telomeric and ribosomal DNA probes were used.

The localization of the ribosomal DNA probe is a standard part of karyotype characterization by molecular cytogenetic methods. For some species, for example, Apodemus, localization of rDNA clusters is species-specific and allows distinguishing different species at the karyotypic level. The identification of interstitial telomeric sites may indicate the presence of recent evolutionary fusions of chromosomes, as was shown in this work.

3) The sequencing of dissected probes is a very nice addition but not clearly described. The authors refer to a methods chapter, but it would still be necessarily to give a very brief description of the material used, the dissection method (how were the chromosomes recognized),

GTG-banding was performed before microdissection to accurately identify chromosomes. This information was added to the text (line 156).

and if there was any further purification before proceeding with the WGA.

No, amplification was carried out in accordance with the manufacturer's protocol. But after the amplification, DNA was purified using Nucleic acid purification kits for DNA (BioSilica).

In addition, I presume that the limited depth of sequencing (coverage was not reported) precludes identification of genes? Please explain why there is no information on numbers of genes found, and on whether known critical genes involved in sex determination are present etc? Will the sequencing data be deposited and accessible to other researchers? I would also appreciate more explanation of the implications of the sequencing results. For example, the neoY appears to cover a region syntenic to mouse chromosome 18. This region is also present on X1 and X2, so if there is a dominant sex determining gene, it should be localized in this region. Perhaps this can be explained.

Unfortunately, there is no way to calculate the genome coverage with our sequencing data, since WGA leads to patchy chromosome coverage, i.e. we are sequencing amplicons, but not full chromosome sequence. In addition, there is a large number of contamination in the reads - DNA sequences of bacteria, humans, and bacterial vectors that occur in WGA kit (mapping statistics are shown in Table S3).

Because we use ultra-low coverage sequencing, we cannot detect L. m. vinogradovi genes, only identify respective orthologous regions in mouse genome. However this information is sufficient to describe the evolutionary origin of the large genomic regions.

Sequencing data is uploaded to the public SRA database, accession number have been added in section 2.8.

Minor comments:

General: your sequencing data nicely show that almost the complete mouse X chromosome is still present on neoX1 and X2. However, these chromosomes also now include “autosomal” parts.

What is known about dosage compensation mechanisms in these animals, is only part of the X heterochromatic/barr body-like in females?

Unfortunately, the mechanism of dose compensation in individuals of this species is still unknown. Forthcoming studies may shed light on this interesting issue.

Is X1 preferentially inactivated and could the relative infrequent occurrence of the type IV karyotype be related to lethality associated with inactivating both X chromosomes? The authors do not need to go into this, but I just wondered what is known about this and whether their data shed any light on dosage compensation mechanisms in these animals.

Unfortunately, studies on the inactivation of X chromosomes in L. m. vinogradovi have not been published yet. In present work we did not perform any inactivation estimates

Lines 244-246:” The autosomes homologous to MAG 17/19 (neo-X3) should also be included in the complex of male sex chromosomes as their homologs were found unpaired in KII.”

The sentence is confusing and the term unpaired is confusing, do the authors mean that the neoX3 is present in single copy?

The sentence was corrected (lines 247-249): “The autosomes homologous to MAG17/19 (neo-X3) should also be included in the complex of male sex chromosomes as they were present in single copy in KII”.

Fig 1: Please indicate KI, KII and KIII in the figure with the appropriate karyotype.

We marked different karyomorphs in the Figure 1.

Also, why are results from DTO probes only shown in b?

Localization of the full set of D. torquatus probes is indeed shown for only one individual. Localization of the same probes on the chromosomes of other individuals was not significantly different, so we decided not to include it. In the text, an additional emphasis was placed on this.

Also please clearly refer to the figure 2b where DTO probe results are mentioned in the main text?

Apparently, you were referring to Figure 1b. We added some information in the beginning of section 3.4. (lines 232-239): “Comparative chromosome painting with two sets of painting probes was used for the analysis of karyotypes of twelve animals (Table 2). Since the set of M. agrestis probes showed almost complete identity of the autosomal sets in various individuals of L. m. vinogradovi, only partial localization of the D. torquatus probes was carried out on the chromosomes of most individuals. Application of comparative chromosome painting allowed us to establish that...”

Fig 3 legend is unclear “females of one type” females of another type” Please indicate which type. The phrasing also suggests that results from only 3 animals are shown, whereas multiple pairs of chromosome 2 are shown in each of the three columns. So please explain more clearly what is shown.

Figure 3 and its legend (lines 266-271) were corrected to make them more readable.

Lines 276-278:” acrocentrics homologous to MAG17/19 (neo-X3). MAGY probe labeled heterochromatic, C-positive parts of neo-X1 and neo-X2 chromosomes of male and female L. m. vinogradovi”

Where is this result shown? There is no Y indicated in the lines representing the MAG chromosomes in Fig 1?

We noted the fact of such localization only in the text, since this signal does not correspond to the real homology of the Y chromosome. We discuss this in section 4.2.

The list of probes in TableS2 does not correspond to the probes listed in table S3.

As L9 library was not used in this study, we deleted information about it form Table S3.

In addition, it is not clear why samples are labelled “double” L9/L9, in addition, the significance of adding “new” is not clear.

These were technical values, we corrected the table deleting double names.

The Page name is in Russian. Please be more precise.

We double checked and did not find the page name…

Line 300-301: this is not very clearly phrased. The sequencing results indicate synteny of the neo-Y chromosome with a region of mouse chromosome 18.

We rephrased the sentence (lines 304-306): “The sequencing data did not show homology of the chromosome to the mouse X chromosome but indicated the synteny of the neo-Y chromosome with a region of the mouse chromosome 18 only”.

Lines 347-352: “The sequencing data confirmed the comparative chromosome painting data showing that MAG13=MMU18, MAG17=MMU13/15, MAG19=MMU15 [26] with one exception. The false-positive was detected by FISH on the neo-Y chromosome. As the sequencing data did not show homology of this chromosome to the mouse X chromosome, we supposed that the detection of the region was caused by FISH-based localization of shared repetitive sequences, while applied here the bioinformatic 351 analysis involved only unique, unrepeated sequence.”

This is unclear, perhaps rephrase along these lines:

Comparing our sex chromosome sequencing data to previous comparative chromosome painting data [26] confirms that MAG13=MMU18, MAG17=MMU13/15, and MAG19=MMU15. However, our neoY sequencing data did not show homology of this chromosome to the mouse X chromosome, and this contrast to the clear detection of X chromosome signal on the neo-Y chromosome by FISH. Therefore, it is likely that the FISH-signal represents shared repetitive sequences that are not included in the bioinformatic analyses.

We replaced this part of the text with the version that was proposed by the reviewer (lines 364-372).

Reviewer 2 Report

The aim of the manuscript by Svetlana A. Romanenko et al., is the analysis of sex determination in the mandarin vole, Lasiopodomys mandarinus. The topic of this research is interesting because of the fascinating sex chromosome polymorphism in this species and many unanswered questions in this area. The authors attempt to solve some problems connected with the evolution of this sex determination system, for example what chromosome rearrangements led to the polymorphism of sex chromosomes. Furthermore, they try to propose a mechanism of sex determination in this species. The authors based their analysis on extensive karyotyping, crossbreeding experiments, molecular cytogenetic methods, and single chromosome DNA sequencing.

In my opinion the work is well written and used tools are proper. However, sometimes the interpretation of results should be clarified better. The results give some important remarks on the sex determination in rodents and underline the need for accurate analysis of investigated Lasiopodomys mandarinus for a better understanding of significant plasticity of its genome, as well as ongoing processes of karyotypic evolution within this species before we try to draw conclusions about a final mechanism of sex determination.

Specific remarks:

1. l. 144, p. 4: usedfor -> used for

2. There is no clear the way of statistical analysis for data in Table 1. For example, if there is ê­•2 test used for KII the expected value should be 149/2=74.5 and ê­•2 value should be 11.28 (because of (54-74.5)2/74.5 + (95-74.5)2/74.5). Based on this p should be about 0.017. I also raise my doubts about the statistical assessment of proportions of KII : KIII : KIV daughters in progeny by ê­•2 test. Could you explain your methodology?

3. l. 320 or 323, p. 10: the usage of the name Y in this places in contrary to neo-Y in other places should be clarified.

4. l. 397, p. 12: The authors claim that sex determination system proposed for L. mandarinus vinogradovi is similar to Myopus schisticolor. In my opinion it would be interesting to describe this comparison in more details. If it is possible add a comment to the statement that the X*Y M. schisticolor females can only produce female offspring. Here, KIII females can produce females as well as males. Moreover, if the disturbed balance between females and males noticed in the analysis is typical of a wild population, what explanation can be for this effect in nature.

5. How can you explain the absence of females among progeny of KIV females in the context of possible mechanisms for b and c examples (Fig.4)? Do you expect females in the progeny of KIV females if you could analyze a bigger-sized sample? Is there any explanation for the smallest reproductive success for KIV females in comparison to predicted?

Author Response

1. 144, p. 4: usedfor -> used for

The part of the text included the typo was rewritten.

2. There is no clear the way of statistical analysis for data in Table 1. For example, if there is ê­•2 test used for KII the expected value should be 149/2=74.5 and ê­•2 value should be 11.28 (because of (54-74.5)2/74.5 + (95-74.5)2/74.5). Based on this p should be about 0.017. I also raise my doubts about the statistical assessment of proportions of KII : KIII : KIV daughters in progeny by ê­•2 test. Could you explain your methodology?

In the previous version of the MS we misrepresented the results of a chi-square test of independence instead of goodness-of-fit test, and the results of goodness-of-fit test instead of the Fisher’s exact test. Both mistakes are corrected (lines 137-141), thank you!

3. l. 320 or 323, p. 10: the usage of the name Y in this places in contrary to neo-Y in other places should be clarified.

In the both cases (lines 323 and 328) we added “neo-”.

4. l. 397, p. 12: The authors claim that sex determination system proposed for L. mandarinus vinogradoviis similar to Myopus schisticolor. In my opinion it would be interesting to describe this comparison in more details. If it is possible add a comment to the statement that the X*Y M. schisticolorfemales can only produce female offspring. Here, KIII females can produce females as well as males. Moreover, if the disturbed balance between females and males noticed in the analysis is typical of a wild population, what explanation can be for this effect in nature.

In Myopus schisticolor, the X*Y females almost always produce daughters only because only X* eggs are formed at mitotic anaphase in their fetal ovaries due to a mechanism of double non-disjunction (Fredga, 1988). In Dicrostonyx, as in the mandarin vole, the X* Y females produce both X*-and Y-carrying eggs. On the other hand, the specific for each female genotype sex ratios of the collared lemmings, unlike these of the mandarin vole, are probably in accordance with the expected proportions – 0.5 for XX, 0.33 for X*Y, and 0.25 for XX* (Gileva, 1998).

In the revised MS, we included a discussion of a sex ratio distortion and non-random chromosome segregation in L. m. vinogradovi into section 4.1. (see below). However, the mention of the lemmings’ sex ratio is removed from section 4.3. of new version of the article. We preferred not to touch upon issues related to the regulation of the sex ratio of other species with aberrant sex determination systems, or the possible population-related effects of a sex ratio distortion in the mandarin vole. To make sense, such discussions require a rather detailed presentation of (in places controversial) information, which, in our opinion, is unnecessary in this already loaded with heterogenous data article.

The unequal proportions of different karyomorphs in progeny of each type of females may be explained by two phenomena. The first one is apparent lower viability of neo-X1/neo-X1 females due to a violation in gene dosage compensation, or some other unknown causes. Unfortunately, the mechanisms of gene dosage compensation in this species are unexplored. The second plausible phenomenon is that the relative success of male neo-X1 and neo-Y spermatozoa depends on the karyomorph of a fertilised female. From our chromosome segregation data, it looks like male gametes carrying neo-Y are favored in neo-X1/neo-X2 females resulting in higher than expected proportions of both sons (neo-X1/neo-Y) and neo-X2/neo-Y daughters (each about 40% vs the expected 25%). To the contrast, an excess of sons (40% vs 33% expected) and neo-X1/neo-X2 females (40% vs 33% expected) in a progeny of neo-X2/neo-Y dams, and predominance of sons in a progeny of neo-X1/neo-X1 dams suggests the male gametes with neo-X1 chromosome to be favored in these cross-breeding combinations. The cytogenetic mechanisms underlying these phenomena are unclear but it is noteworthy that they all reduce a sex ratio distortion. Thus, in terms of ultimate causes, these mechanisms might be selected as they increased the investment to sons which have higher reproductive value than daughters in a population with a female-biased primary sex ratio (Fisher, 1930).

5. How can you explain the absence of females among progeny of KIV females in the context of possible mechanisms for b and c examples (Fig.4)? Do you expect females in the progeny of KIV females if you could analyze a bigger-sized sample? Is there any explanation for the smallest reproductive success for KIV females in comparison to predicted?

Theoretically, in the progeny of females of KIV, the ratio of sons and daughters should be the same. However, we observed, firstly, a general decrease in fertility in KIV females, and, secondly, the absence of females among their offspring. It is possible that with a significant increase in sampling in the offspring of KIV females, females will be detected. Since we do not know anything about the mechanisms of dose compensation in this species, we can only assume that a violation of the dose of the genes can lead to poor survival of female KIV. The corresponding sentences have been added to the section 4.1. (see above).

Round 2

Reviewer 2 Report

The authors tried to improve their manuscript. They added some new comments and removed some information that could have cause a confusion in the previous version. In my opinion the main subject is now better presented but not in all parts. Moreover, I noticed some problems in the second version. Thus, the specific comments are listed below.

l. 360, p. 8: The status of the Figure 2 is not understood. Does this Figure stay? It looks like removed.

l. 462-464, p. 10: There is a repetition in two sentences.

l. 543, p. 11: The authors mention locus E in description Figure 4a whereas this locus is introduced only in Figure 4c.

l. 551, p. 12: The sign © should be changed for (C).

l. 551-552, p. 12: The Figure 4b was not changed but the description in text yes. Did the authors resign from the role of b gene or C* allele? The caption of the Figure should be consistent with the text.

l. 556-562, p.12 Here, I assume that the description is for Figure 4a rather, not for Figure 4c. It should be clarified.

The text and the Figure 4 demand corrections before publication. The possible mechanisms of sex determination should be consistent with the Figure 4.

l. 564, p.12: The wrong place for citations [41][42][43][8].